# Spatiotemporal Variation and Driving Factors Analysis of Habitat Quality: A Case Study in Harbin, China

Yuxin Qi [1] and Yuandong Hu [1,2,*]

1    College of Landscape Architecture, Northeast Forestry University, Harbin 150040, China;
     qiyuxin@nefu.edu.cn
2    Institute for Interdisciplinary and Innovation Research, Xi'an University of Architecture and Technology,
     Xi'an 710055, China
*    Correspondence: huyuandong@nefu.edu.cn; Tel.: +86-451-82190492

**Abstract:** Biodiversity is profoundly influenced by habitat quality, and Harbin, a provincial capital situated in a cold climate zone, stands out as one of China's regions most susceptible to the repercussions of climate change. To ensure the city's continued sustainable growth, a thorough assessment of habitat quality must be conducted. This study employs a comprehensive approach integrating the InVEST model, the PLUS model, a landscape pattern analysis, geographic detector, and a geographically weighted regression model. The goal is to assess how land use and habitat quality have changed in Harbin City, investigate factors contributing to spatial heterogeneity in habitat quality, thoroughly examine evolutionary patterns under the inertial development scenario from 2030 to 2050, and propose spatial optimization strategies. There are four key findings. First, from 2000 to 2020, agricultural land and forest were Harbin City's two most prevalent land use types. The most notable transition occurred from forest to grassland, and the expansion of construction land primarily resulted from its encroachment into agricultural areas. Second, within the area of study, the landscape heterogeneity increased while simultaneously experiencing a decrease in connectivity, and the landscape had a tendency toward a more fragmented spatial distribution. Third, overall habitat quality rose between 2000 and 2020 but declined between 2030 and 2050. There was a "weak in the west and high in the east" distribution pattern in the spatial heterogeneity of habitat quality. Fourth, population density has the most impact on habitat quality, with the NDVI and GDP close behind. Conversely, precipitation and slope had comparatively smaller influences on habitat quality. Natural factors combined had a primarily favorable influence on habitat quality across the research region in terms of spatial distribution. Conversely, population density had a discernibly detrimental impact. Given these findings, this study suggests targeted strategies to optimize habitat quality. These recommendations are relevant not only for biodiversity conservation but also for the development of an ecologically sustainable community, particularly in a cold climate region.

**Keywords:** habitat quality; landscape pattern; InVEST model; PLUS model; geographic detector; geographically weighted regression



## 1. Introduction

Biodiversity is the cornerstone of sustainable urban development, and its decline has become a significant global environmental issue [1,2]. Habitat quality is the measure of an ecosystem's ability to furnish an environment conducive to the living and growth of species [3] and is a key indicator of biodiversity status [4,5]. The conservation of biodiversity is seriously threatened by its destruction [6,7].

Changes in land use and cover (LULC) may indicate the extent to which human activity has harmed an ecosystem [8] and are the most direct representation of how human actions interact with the environment [9]. Traditionally, landscape pattern serves as a tool to describe the structural features of LULC, encompassing aspects like shape, proportion,

and complexity. It offers a comprehensive depiction of the ecological environment system within a region [10,11]. Strategic habitat quality optimization requires an understanding of the features of LULC evolution and the distribution of landscape patterns. Widespread human development and building activities have significantly changed the types, patterns, and intensities of LULC in the context of growing urbanization. These changes have resulted in a discernible deterioration in habitat quality [12–14], with serious impacts on biodiversity and human well-being [12]. Given this context, it becomes essential to thoroughly explore the spatiotemporal intricacies and underlying driving forces that give rise to variations in habitat quality. This exploration is imperative not just for biodiversity conservation but also for fostering human well-being [4,15,16].

Habitat quality research has studied macro- and microscales. Generally, habitat quality evaluation methods are categorized into two types: (1) establishing an indicator evaluation system and obtaining habitat parameters through field surveys and (2) evaluating habitat quality using various types of models. While the field survey approach is constrained to specific areas, the advent of various models has empowered scientists to assess habitat quality across expansive regions. The InVEST model in particular features a dedicated habitat quality module crafted to systematically evaluate ecological and environmental conditions by integrating changes in LULC and biodiversity threats [17]. It has been utilized in several global locations, including southwest Ethiopia [18], the state of Georgia in the United States [19], and the Weihe River Basin in China [20]. The model has combined a system dynamics model [20], the PLUS model [21], the coupled coordination degree model [22], and different approaches to investigating future possibilities of habitat quality and its connection to urbanization to serve as a foundation for scientific planning and spatial optimization for various ecosystems, such as national parks [23], watersheds [24], cities [25,26], and coastal zones [27].

Most research methods on habitat quality for large-scale regions focus on analyzing spatiotemporal evolution and future predictions [21,22,28,29]. Despite considerable advances in this area, the exploration of the underlying drivers influencing habitat quality requires further study [30]. Geographic detector (GD), a commonly used statistical method of exploring spatial heterogeneity, can explore not only the impact of each influencing factor on the explained variables but also the interactive explanatory power of multiple factors. However, GD is limited in spatially articulating the magnitude of the impact of the various drivers. In contrast, from a geospatial viewpoint, the geographically weighted regression (GWR) model is excellent for dissecting the effect mechanisms of spatiotemporal differences in habitat quality. Thus, GWR is an important analytical tool to explore the many factors that influence habitat quality across geographical scales [31].

Harbin is a typical provincial capital city in the cold region of China. It is highly sensitive to climate change and experiences more significant changes in the geographic distribution of ecological risk [32]. The current study has the following aims: (1) to use methods such as the LULC transition matrix, landscape pattern analysis, and the InVEST model to quantitatively delineate the spatiotemporal dynamics of Harbin's landscape types, landscape configurations, and habitat quality; (2) to identify the key determinants of Harbin's habitat quality from 2000 to 2020 based on GD; (3) to use GWR to investigate how social and natural factors affect habitat quality in Harbin; and (4) to use the PLUS model to forecast the likely spatial distribution of habitat quality in Harbin from 2030 to 2050. The ecological service assessment and urban planning of Harbin City can be supported scientifically by the findings of this study.

## 2. Materials and Methods

### 2.1. Study Area

Harbin City is in the northeast of China's Northeast Plain, covering an approximate area of 53,000 km$^2$ (Figure 1). As a regional central city in northeastern China and in the first set of national comprehensive pilot cities for new urbanization, the urban agglomeration in the northeast benefits greatly from Harbin's contribution to sustainable development.

Climatically, Harbin is classified as a mid-temperate continental monsoon zone, with year-round precipitation mainly concentrated from June to September, with an average annual precipitation of 539.03 mm and an average annual temperature of 4.9 °C. As a representative provincial capital city in the cold region of Northeast China, Harbin has significant climate change characteristics, fertile agricultural zones, and a large area. It is rich in natural resources, with black soil as the main type of soil and abundant land resources. In recent years, the urbanization level of Harbin City has been advancing, land use has changed significantly, and habitat quality has also changed. Based on this, this study evaluates habitat quality in Harbin City from 2000 to 2020 and simulates the development pattern of habitat quality from 2030 to 2050, which can provide a reference for future urban development.

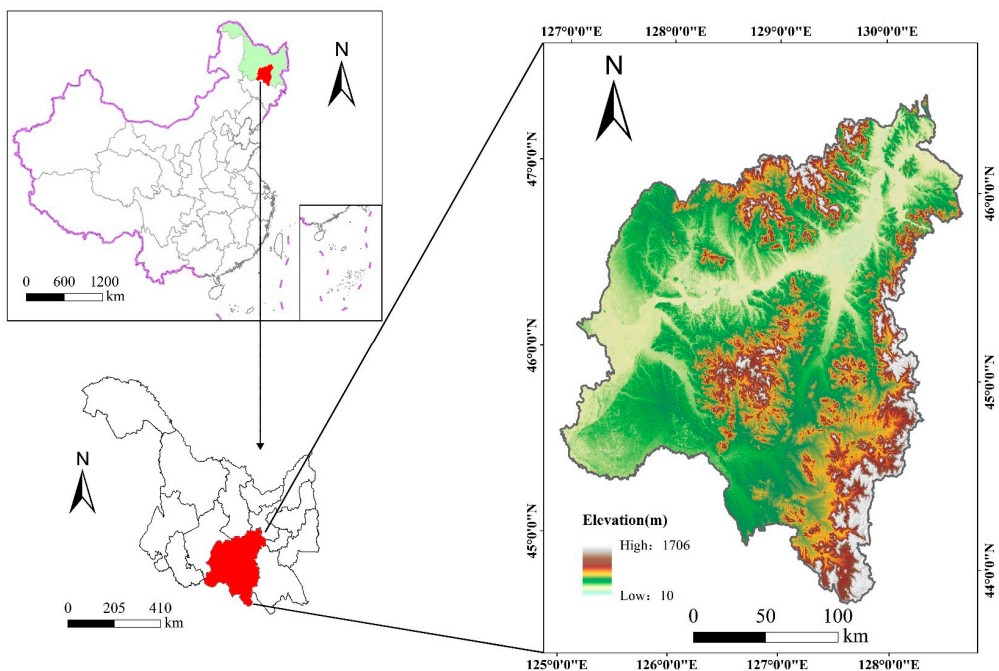

**Figure 1.** Location of Harbin City in northeastern China.

*2.2. Data Sources*

The main data sources used in this study are shown in Table 1.

**Table 1.** Data sources.

| Data | Data Source | Spatial Resolution | Temporal Resolution | Accessed Date |
|---|---|---|---|---|
| Land use/land cover | GlobeLand30 (http://www.globallandcover.com/) | 30 m × 30 m | 2000, 2010, and 2020 | 15 October 2022 |
| Annual average precipitation | National Earth System Science Data Center, National Science & Technology Infrastructure of China (http://www.geodata.cn) | 1 km × 1 km | 2000, 2010, and 2020 | 15 October 2022 |
| Temperature | National Earth System Science Data Center, National Science & Technology Infrastructure of China (http://www.geodata.cn) | 1 km × 1 km | 2000, 2010, and 2020 | 15 October 2022 |

**Table 1.** *Cont.*

| Data | Data Source | Spatial Resolution | Temporal Resolution | Accessed Date |
|---|---|---|---|---|
| DEM | Geospatial data cloud (http://www.cloud.cn) | 30 m × 30 m | 2009 | 6 December 2022 |
| NDVI | The United States Geological Survey (https://lpdaac.usgs.gov/products/mod13q1v061/) | 250 m × 250 m | 2000, 2010, and 2020 | 8 March 2023 |
| Population density data | WorldPop Hub (https://hub.worldpop.org/) | 1000 m × 1000 m | 2000, 2010, and 2020 | 8 March 2023 |
| GDP | China's Resource and Environmental Sciences Data Centre (https://www.resdc.cn/) | 1000 m × 1000 m | 2000, 2010 and 2019 | 8 March 2023 |
| Roads and rivers | National Geographic Information Resource Directory Service System (https://www.webmap.cn/main.do?method=index) | — | — | 8 March 2023 |

*2.3. Methods*

2.3.1. Land Use Transfer Change

The volume and direction of LULC type transfers over the study period can be reflected in the LULC transfer matrix, which is used to characterize transfers between various LULC types [17–19]. The land use transfer matrix is calculated using the following formula:

$$S_{ij} = \begin{bmatrix} S_{11} & S_{12} & \cdots & S_{1n} \\ S_{21} & S_{22} & \cdots & S_{2n} \\ & & \vdots & \\ S_{n1} & S_{n2} & \cdots & S_{nn} \end{bmatrix} \tag{1}$$

where $S$ denotes the area of LULC; $n$ indicates the number of types of LULC; and $S_{ij}$ is the area transferred from LULC type $i$ to type $j$ at the start and finish of a time period.

2.3.2. Landscape Indices

Using FRAGSTATS 4.2, this study computed regional edges, patch form complexity, landscape aggregation, fragmentation, and diversity to measure landscape patterns (Table 2). The LULC data for the calculation of the landscape indices were categorized based on the GlobeLand30 dataset. At both landscape and land use type levels, this study calculated landscape indices for the whole city of Harbin and different land uses. To spatialize the distribution of landscape patterns in Harbin City, this study used the moving window method to calculate the spatial distribution of each index. A moving window scale that is too small leads to discontinuity in the generated image, while a window that is too large leads to a loss of overall image detail and blurring of the generated image. In this study, GS+ 9.0 software was used to simulate the semi-variance function of landscape indices under different moving window radii and to calculate the nugget/sill ratio to determine the optimal moving window size. When the window radius is 720 m, the fluctuation in the landscape index nugget/sill ratio starts to decrease, so 720 m is the optimal moving window radius.

**Table 2.** Descriptions of landscape indices.

| Types | Landscape Indices | Abbreviation |
|---|---|---|
| Area-edge | Largest Patch Index | LPI |
| | Percentage of Landscape | PLAND |
| Shape | Landscape Shape Index | LSI |
| | Aggregation Index | AI |
| Aggregation | Contagion Index | CONTAG |
| Subdivision | Number of Patches | NP |
| | Patch Density | PD |
| | Landscape Division Index | DIVISION |
| Diversity | Shannon's Diversity Index | SHDI |

### 2.3.3. Habitat Quality

The examination of Harbin City's habitat quality spanned from 2000 to 2020 based on the InVEST model. The main formulas for the calculations are given below [20,22]:

$$Q_{xj} = H_j \left[ 1 - \left( \frac{D_{xj}^Z}{D_{xj}^Z + k^z} \right) \right]$$ (2)

$$D_{xj} = \sum_{r=1}^{R} \sum_{y=1}^{Y_r} \left( \frac{w_r}{\sum_{r=1}^{R} W_r} \right) r_y i_{rxy} \beta_x S_{jr}$$ (3)

where $Q_{xj}$ indicates the habitat quality of a LULC type in a grid cell $x$; $H_j$ indicates the habitat suitability of LULC type $j$; $D_{xj}^Z$ is the habitat degradation degree of grid $x$ in land use type $j$; $k$ is the half-saturation constant; $Z$ indicates normalized constants, and $Z = 2.5$ is set as programmed; $x$ is a constant; $y$ indexes all grid cells on $r$'s map and $Y_r$ corresponds to the set of raster cells of $r$'s map; $r_y$ is the intensity of the threat factor; $\beta_x$ is the anti-interference level of habitat; and $S_{jr}$ is the relative sensitivity degree of different habitats to different threat factors.

Based on earlier research and taking into account the real circumstances of the studied region [33,34], habitat quality was categorized into five levels: low (0–0.2), low–medium (0.2–0.4), medium (0.4–0.6), medium–high (0.6–0.8), and high (0.8–1).

### 2.3.4. Spatial Autocorrelation Analysis

A spatial autocorrelation is the correlation of a geographic property over many geographic locations [35,36].

The global spatial autocorrelation was calculated as follows:

$$I = \frac{n \sum_{i=1}^{n} \sum_{j=1}^{n} w_{ij}(x_i - \overline{x})(x_j - \overline{x})}{\sum_{i=1}^{n} \sum_{j=1}^{n} w_{ij} \sum_{i=1}^{n}(x_i - \overline{x})^2}$$ (4)

The calculation formula for local spatial autocorrelation is as follows:

$$I_i = \frac{(x_i - \overline{x})}{S_x^2} \sum_{j=1, j \neq i}^{n} w_{ij}(x_j - \overline{x})$$ (5)

where $I$ is the global spatial autocorrelation index; $I_i$ indicates the local spatial autocorrelation index; $n$ indicates the number of regions; $x_i$ and $x_j$ are the index values of samples $i$ and $j$, respectively; $\overline{x}$ denotes the average sample index; $w_{ij}$ is the spatial relationship weight matrix; and $S_x^2$ denotes the variance in the observation unit $x_j$.

2.3.5. Geographic Detector

The determinants of the geographical variability in the habitat quality distribution in Harbin City were identified via the application of factor detection and interaction detection in geographic detectors (GDs). Higher values suggest a better ability to explain the regional variation in habitat quality, with a q-value ranging from 0 to 1. The factor detection primarily evaluates the capacity of various drivers in the research area for habitat quality [37]. The formula was calculated as follows:

$$q = 1 - \frac{\sum_{h=1}^{L} N_h \sigma_h^2}{N \sigma^2} = 1 - \frac{SSW}{SST} \tag{6}$$

$$SSW = \sum_{h=1}^{L} N_h \sigma_h^2, SST = N\sigma^2 \tag{7}$$

where $h$ ($h$ = 1, 2, ..., L) indicates the stratification of variable Y or factor X; $N_h$ and $N$ indicate the number of cells in stratum $h$ and the whole region, respectively; $\sigma_h^2$ and $\sigma^2$ are the variance of the $Y$ values in stratum $h$ and the whole region, respectively; and $SSW$ and $SST$ denote the sum of the variances within the stratum and the total variance in the whole study area, respectively.

Interaction detection was employed to evaluate whether the combined action of the two drivers either heightened or lessened the explanatory power of habitat quality.

2.3.6. Geographically Weighted Regression

To better account for regional heterogeneity, GWR develops localized coefficients by using spatial location characteristics [38,39]. The calculation formula was

$$y_k = \beta_0(u_k, v_k) + \sum_{i=1}^{n} \beta_i(u_k, v_k) x_{ki} + c_k \tag{8}$$

where $y_k$ denotes the ESs value; $x_{ki}$ is the landscape index; $n$ is the total number of spatial units involved in the analysis; $c_k$ is the random error term; $(u_k, v_k)$ is the spatial location of sample $k$; $\beta_0(u_k, v_k)$ is the intercept at location $k$; and $\beta_i(u_k, v_k)$ is the coefficient of the $i$-th independent variable of sample $k$.

2.3.7. Simulation and Prediction of LULC Based on PLUS Model

The PLUS model introduces a framework that relies on land expansion analysis techniques and a cellular automata model featuring various stochastic seeds. This approach effectively reveals the fundamental drivers behind changing landscapes and the expansion of land use. Comparatively speaking, the PLUS model offers superior simulation accuracy [40]. Based on previous studies and data availability, 10 factors were selected as drivers of LULC change from both natural and social aspects (Table 3) [21,41,42]. The PLUS model LEAS (land expansion analysis strategy) module automatically calculates the extent to which all drivers contribute to the expansion of each land use type.

This research assessed the changes in habitat quality in Harbin City under the inertial development scenario from 2030 to 2050, using a 10-year span. In this scenario, land use types spontaneously increase in diverse geographical and temporal dimensions; hence, there are no areas of restriction on transfers. The transfers between land uses for the future scenario are based on the actual land use transfers from 2010 to 2020 to set the land use transfer matrix.

In evaluating the accuracy of the PLUS model, this study employed the Kappa coefficient. A comparison was made between the actual 2020 LULC data and the simulated 2020 LULC results, generated based on data spanning from 2010 to 2020. The accuracy obtained overall was 0.89, with a Kappa coefficient of 0.83, indicating a high degree of consistency between the two datasets. These results affirm that the model's accuracy meets the criteria set forth in this study.

**Table 3.** The spatial parameters that drive the shift in LULC in this research.

| Category | Driving Factors |
|---|---|
| Natural factors | Average annual precipitation (PRE)<br>Average annual temperature (TEM)<br>Elevation (DEM)<br>Slope (SLO)<br>Normalized difference vegetation index (NDVI)<br>Distance from water |
| Social factors | Population (POP)<br>Gross domestic product (GDP)<br>Distance from railways<br>Distance from highways |

## 3. Results

### 3.1. Landscape Pattern Changes from 2000 to 2020

3.1.1. Spatial and Temporal Changes in LULC

Figure 2 illustrates the considerable structural complexity of and variability in the LULC types in Harbin City from 2000 to 2020. LULC types were dominated by agricultural land and forest: agricultural land was nearly 50% of the study area, forest was more than 35%, and the total area of the two was more than 80%. There was around 8% grassland in the total area. Wetlands, water bodies, construction land, and bare land had smaller proportions, with a total share of less than 8%. Over these 20 years, land conversions occurred between almost every pair of LULC types. The predominant conversion was from forest to grassland. Specifically, this conversion involved 694.83 km² of forest from 2000 to 2010 and 872.70 km² from 2010 to 2020. From 2000 to 2020, a total of 999.92 km² of forest was turned into grassland. Subsequently, the transformation of grassland into forest (832.77 km²), the transition from agricultural land to forest cover (796.67 km²), and the shift from agricultural land to construction zones (772.89 km²) underscored the predominant source of urban expansion as encroachment upon agricultural lands.

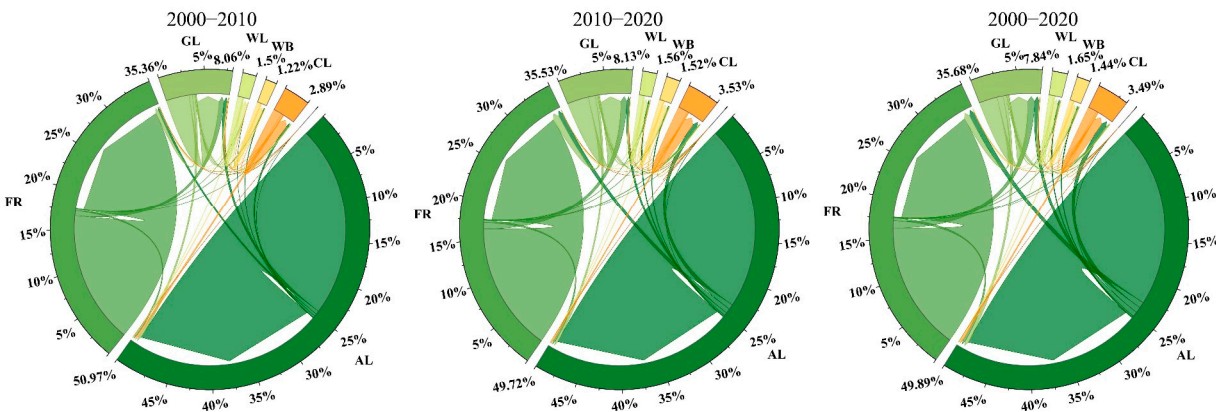

**Figure 2.** LULC change chord diagram, 2000, 2010, and 2020. AL: agricultural land; FR: forest; GL: grassland; WL: wetland; WB: water body; CL: construction land.

3.1.2. Analysis of the Evolution of Landscape Pattern Features

Changes in the Harbin City landscape index at the landscape level are significant from 2000 to 2020, and the results are shown in Table 4. Overall, the number of patches, the patch density, the largest patch index, and the contagion index showed a continuous decrease, and Shannon's diversity index and landscape division index showed a continuous increase. The aggregation index had a lowering and then increasing trend, whereas the landscape shape index displayed a rising and then falling trend. The overall fragmentation of the city decreased, and human interference continued to decrease, while discrete land-

scape continued to increase, landscape connectivity decreased, and spatial heterogeneity increased.

**Table 4.** Variations in landscape indices at the landscape level from 2000 to 2020.

| Year | NP | PD | LPI | LSI | SHDI | CONTAG | DIVISION | AI |
|------|------|------|------|------|------|--------|----------|------|
| 2000 | 259,906 | 4.90 | 28.70 | 197.18 | 1.13 | 65.34 | 0.8764 | 94.92 |
| 2010 | 253,173 | 4.77 | 28.55 | 199.41 | 1.14 | 65.04 | 0.8778 | 94.86 |
| 2020 | 236,092 | 4.45 | 26.99 | 195.49 | 1.19 | 63.71 | 0.8854 | 94.97 |

Between 2000 and 2020, there were significant changes in the landscape pattern index for each LULC type, as depicted in Figure 3. Due to its small area, bare land is not discussed. The number of patches and patch density for agricultural land, grassland, and water bodies exhibited a consistent decrease, suggesting a reduction in fragmentation and an overall trend toward concentration. The number of patches and the patch density of the forest exhibited an initial increase followed by a subsequent decrease. This pattern suggests that landscape fragmentation underwent a transitional process characterized by a shift from "concentration" to "dispersion" and back to "concentration". In contrast, the number of patches and the patch density of wetlands both exhibited an initial decline, followed by a subsequent upward trend, while construction land experienced a consistent and uninterrupted increase. This pattern suggests a continuous escalation in the level of landscape fragmentation. The largest patch index of agricultural land was the highest, indicating that agricultural land was the dominant patch type in Harbin, with a relatively high degree of internal connectivity and a more concentrated landscape, which also reflected how heavily human activity had impacted agricultural land. The landscape shape index consistently exhibited the highest values for grassland, followed by forest and construction land. This pattern signifies that the landscape configuration of grassland was more intricate, while agricultural land, water bodies, and wetlands displayed comparatively smaller landscape shape index values. Agricultural land always had the highest percentage of landscape, although overall the trend was downward, with the forest showing a tendency toward decline followed by an increase and the grassland showing a trend toward gain followed by a decrease. The percentage of the landscape of construction land continued to increase, and the degree of this increase was more drastic from 2010 to 2020, indicating that the urban expansion rate was faster and the urbanization level was higher during this period.

3.1.3. Landscape Index Analysis Based on Moving Window Method

Figure 4 displays the geographical distribution of landscape patterns in the research area from 2000 to 2020, with notable overall changes. In the western part of the city, the number of patches, patch density, and landscape shape index were located in low-value areas, but the largest patch index and aggregation index were located in high-value areas. The majority of LULC in Harbin was agricultural land, which comprised almost half of the city's total area. It included high levels of patch connectedness, high levels of aggregation, low levels of landscape fragmentation, and high levels of anthropogenic disturbance. The northern part of the city is the Xiaoxing'an Mountains ecological barrier area, which has low patch number, patch density, Shannon's diversity index, and landscape division index values, as well as high contagion index values. It has a low degree of fragmentation, good connectivity, low landscape heterogeneity, and a good ecological environment. The regional economy's high-quality development area, where the city center is located, has a comparatively low contagion index and a high patch number, patch density, and landscape shape index. These data indicate the area's high degree of landscape fragmentation and the strength of human interference activities.

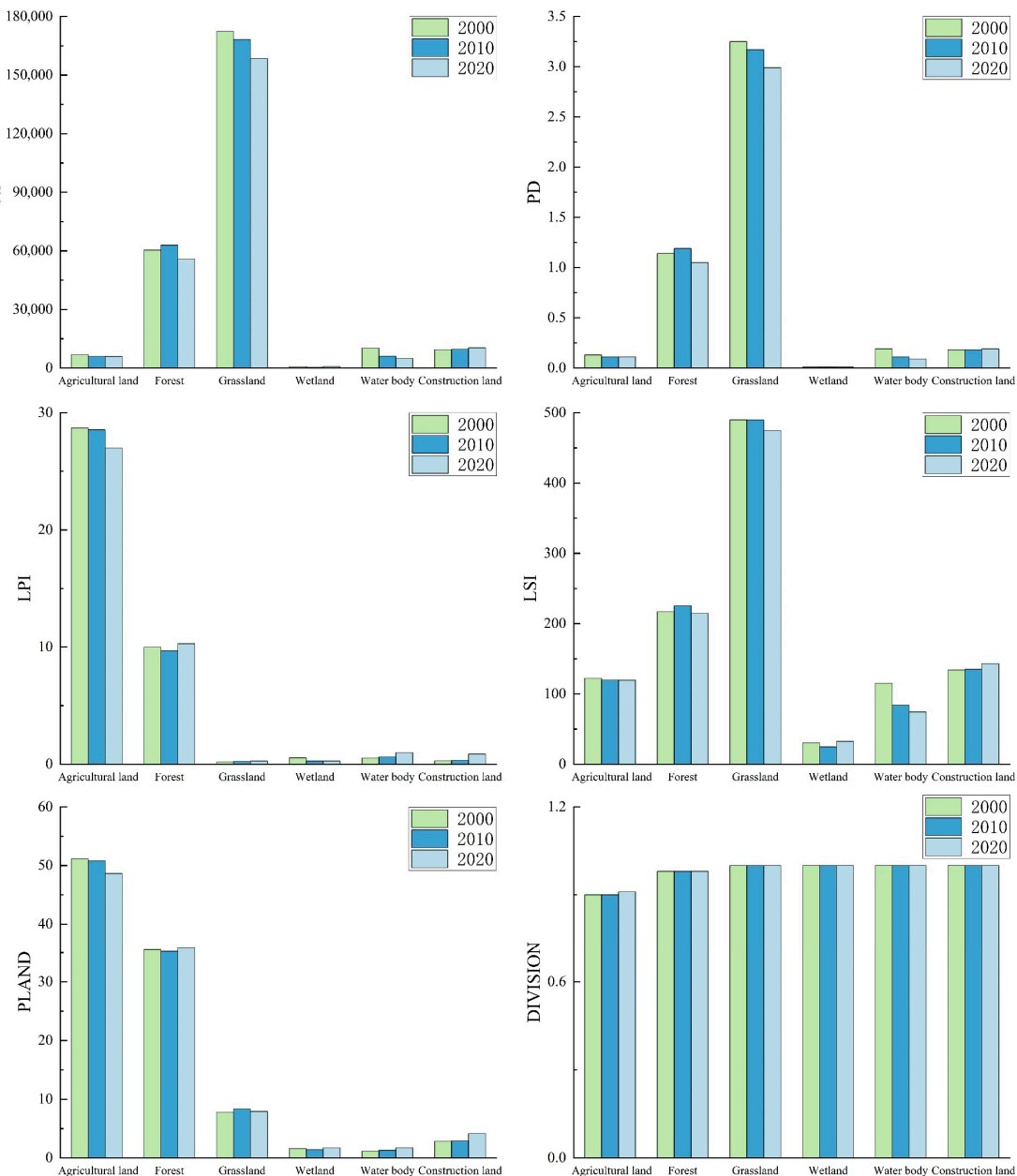

**Figure 3.** Landscape pattern index at the class level from 2000 to 2020.

### 3.2. Habitat Quality Changes

3.2.1. Spatial and Temporal Evolution of Habitat Quality

Tables 5 and 6 provide the area proportions and transfer matrices for each grade of habitat quality. Importantly, there has been a discernible enhancement in the overall habitat quality of Harbin over time. Analyzing habitat quality ratings, it can be observed that the majority of the city's territory consistently belonged to the medium-grade habitat category, accounting for nearly 50% of the total area. However, this percentage slightly decreased from 52.50% in 2000 to 48.25% in 2020. The LULC type of this medium-quality habitat is mainly agricultural land. Every year, Harbin City's agricultural land area shrinks, and the percentage of medium-quality habitat area also keeps declining. Approximately 40% of the area was a high-quality habitat, and this percentage rose with time.

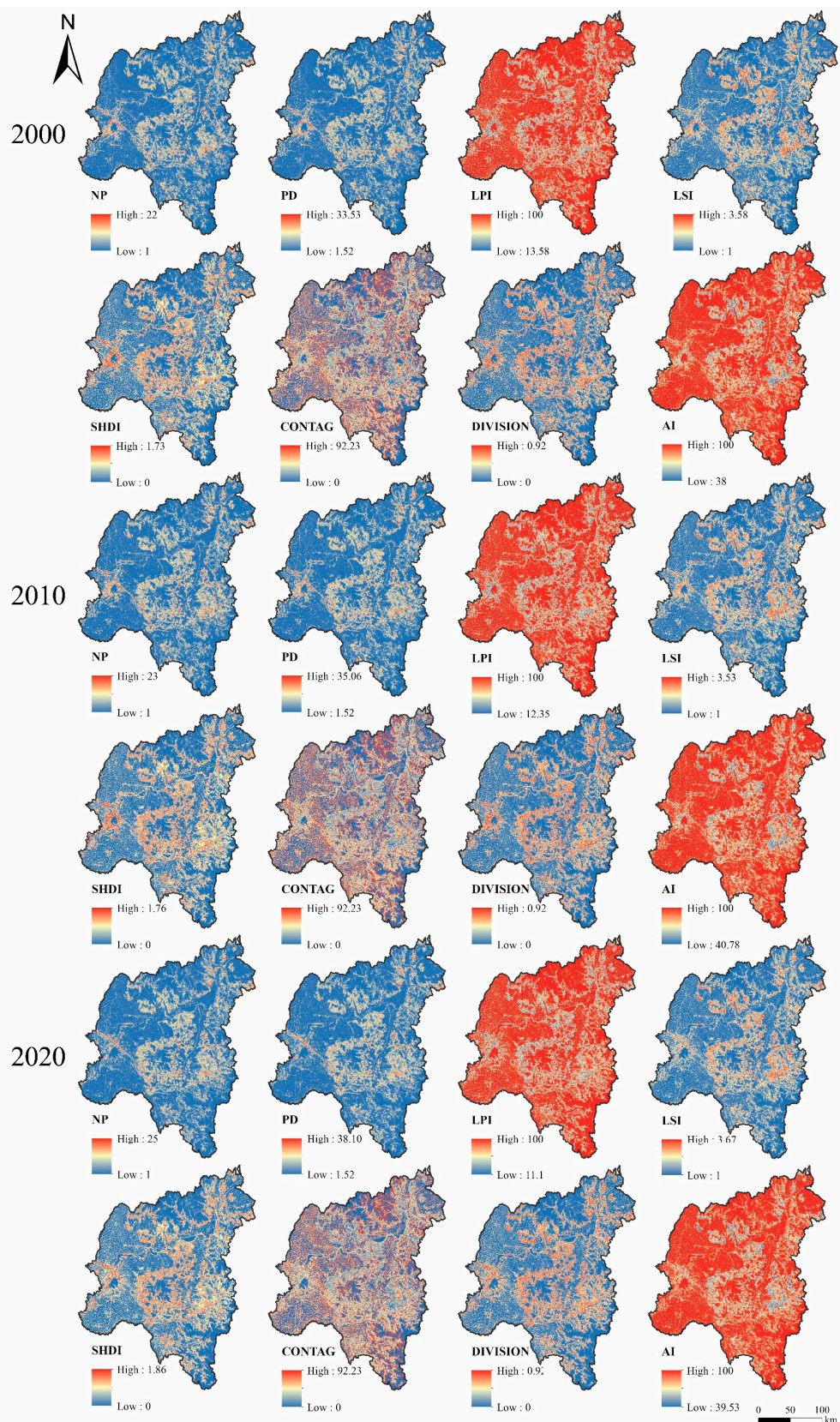

**Figure 4.** The landscape indicators' spatial distributions at the landscape level from 2000 to 2020.

**Table 5.** The area and proportion of habitat quality at different levels from 2000 to 2020.

| Levels | 2000 | | 2010 | | 2020 | |
|---|---|---|---|---|---|---|
| | Area/km$^2$ | Proportion/% | Area/km$^2$ | Proportion/% | Area/km$^2$ | Proportion/% |
| Low | 1516.51 | 2.86 | 1562.49 | 2.94 | 2245.94 | 4.23 |
| Low-medium | 528.921 | 1.00 | 527.631 | 0.99 | 707.73 | 1.33 |
| Medium | 27,853.6 | 52.50 | 27,489.9 | 51.79 | 25,603.9 | 48.25 |
| Medium-high | 2224.48 | 4.19 | 2025.23 | 3.82 | 1561.73 | 2.94 |
| High | 20,934.5 | 39.46 | 21,470.3 | 40.45 | 22,943.8 | 43.24 |

**Table 6.** Transfer matrix of habitat quality in Harbin from 2000 to 2020 (km$^2$).

| Year | Levels | Low | Low–medium | Medium | Medium–High | High | Total |
|---|---|---|---|---|---|---|---|
| 2000–2010 | Low | 1177.84 | 15.97 | 259.25 | 21.40 | 36.75 | 1511.21 |
| | Low–medium | 18.49 | 380.32 | 99.68 | 16.03 | 12.14 | 526.67 |
| | Medium | 261.89 | 111.13 | 26,534.41 | 326.62 | 649.13 | 27,883.19 |
| | Medium–high | 70.62 | 14.62 | 157.66 | 1507.59 | 468.73 | 2219.23 |
| | High | 28.89 | 4.93 | 462.64 | 146.57 | 20,273.54 | 20,916.57 |
| 2010–2020 | Low | 1377.26 | 11.01 | 154.29 | 4.12 | 11.06 | 1557.74 |
| | Low–medium | 167.45 | 216.84 | 111.53 | 17.82 | 13.33 | 526.97 |
| | Medium | 594.98 | 455.27 | 24,760.49 | 663.90 | 1040.18 | 27,514.83 |
| | Medium–high | 59.30 | 21.01 | 141.18 | 665.58 | 1131.41 | 2018.48 |
| | High | 41.36 | 4.34 | 456.43 | 205.27 | 20,736.30 | 21,443.70 |
| 2000–2020 | Low | 1219.95 | 17.93 | 234.59 | 5.13 | 33.61 | 1511.21 |
| | Low–medium | 155.35 | 163.32 | 169.33 | 19.48 | 19.18 | 526.67 |
| | Medium | 713.68 | 504.65 | 24,424.20 | 818.90 | 1421.64 | 27,883.06 |
| | Medium–high | 103.13 | 16.87 | 203.74 | 499.95 | 1395.41 | 2219.11 |
| | High | 48.24 | 5.69 | 590.76 | 213.03 | 20,058.73 | 20,916.46 |
| | Total | 2240.35 | 708.46 | 25,622.62 | 1556.49 | 22,928.57 | 53,056.51 |

In terms of habitat quality transfer matrix, 649.13 km$^2$ of medium-quality habitat was converted into high-quality habitat from 2000 to 2010, and 1040.18 km$^2$ was converted from 2010 to 2020, totaling 1421.64 km$^2$ over the 20 years, followed by 1395.41 km$^2$ of medium-high quality habitat upgraded to high-quality habitat over 20 years. An area of 818.90 km$^2$ of medium-quality habitat was converted into medium–high-quality habitat, and 713.58 km$^2$ was converted into low–medium-quality habitat. In summary, there is an overall improvement in habitat quality, with the high-quality habitat area expanding, but the area of low-quality habitat is also increasing.

As seen in Figure 5, there were notable geographic variations in habitat quality, with a pronounced "poor in the west and high in the east" trend. Differential degrees of urbanization and human activity, which were more prominent in the western city area, were strongly associated with this distribution. The western part of the city, predominantly consisting of construction and agricultural land, exhibited a prevalence of low- and medium-quality habitats. Notably, low-quality habitats were expanding in this area. In contrast, the northern Xiaoxing'an Mountains ecological barrier and the central and eastern portions of the city within the Zhangguangcai Range ecological barrier offered high-quality habitats due to forest-dominated land use and less human disturbance, creating a relatively healthier ecological environment.

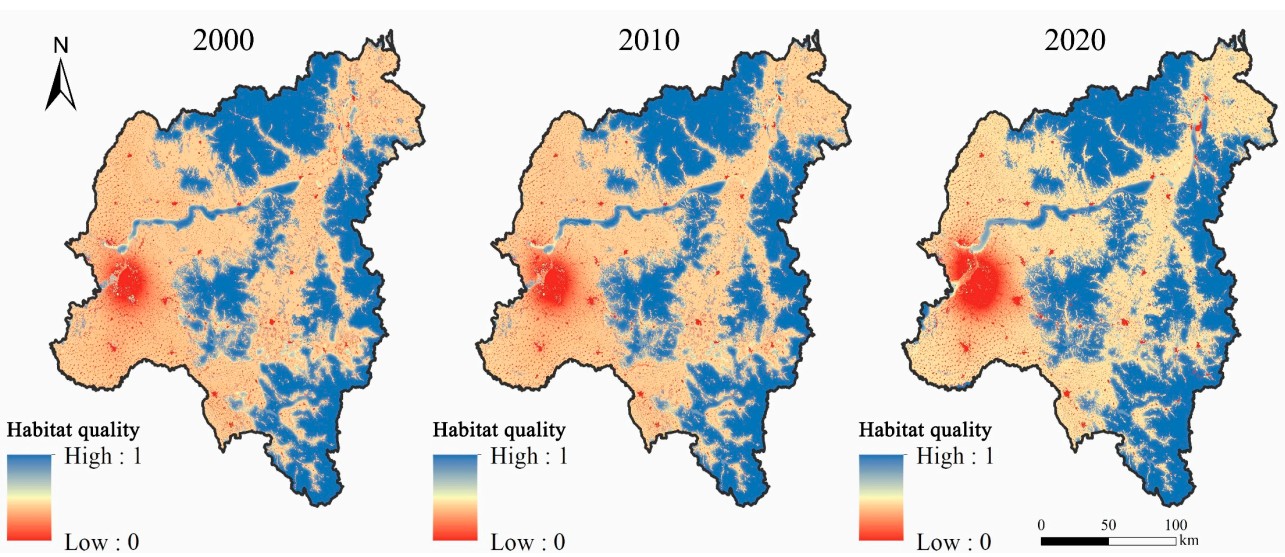

**Figure 5.** Spatial distribution pattern of habitat quality from 2000 to 2020.

### 3.2.2. Spatial Autocorrelation Analysis of Habitat Quality

The global Moran's I index exhibited values of 0.903 in 2000, 0.905 in 2010, and 0.903 in 2020, with a *p*-value of 0.001. This indicates a substantial positive spatial association in the distribution of habitat quality patterns in Harbin City, as seen in Figure 6 on a local spatial autocorrelation. The regions identified to have low–low habitat quality were mainly concentrated in the western part of the city, displaying a more evident degree of aggregation. The two types of land use that dominated these areas were agricultural and building land. The low–low area comprised 21.21% of the city's area in 2000 but declined to 16.53% in 2020, demonstrating that the area of low habitat quality is decreasing. High–high areas increased from 27.11% of the city's total area in 2000 to 28.53% in 2020; they are mainly distributed in the northern, central, and eastern parts of the city and are the most obvious in the Xiaoxing'an Mountains ecological barrier area and the Zhangguangcai Range ecological barrier area in large contiguous areas with forest as the dominant LULC type. Low–high and high–low types accounted for a relatively small proportion, and there was no obvious centralized distribution. Overall, the level of urbanization in Harbin City has been increasing, the level of habitat quality has also increased, and the ecological environment has been improving.

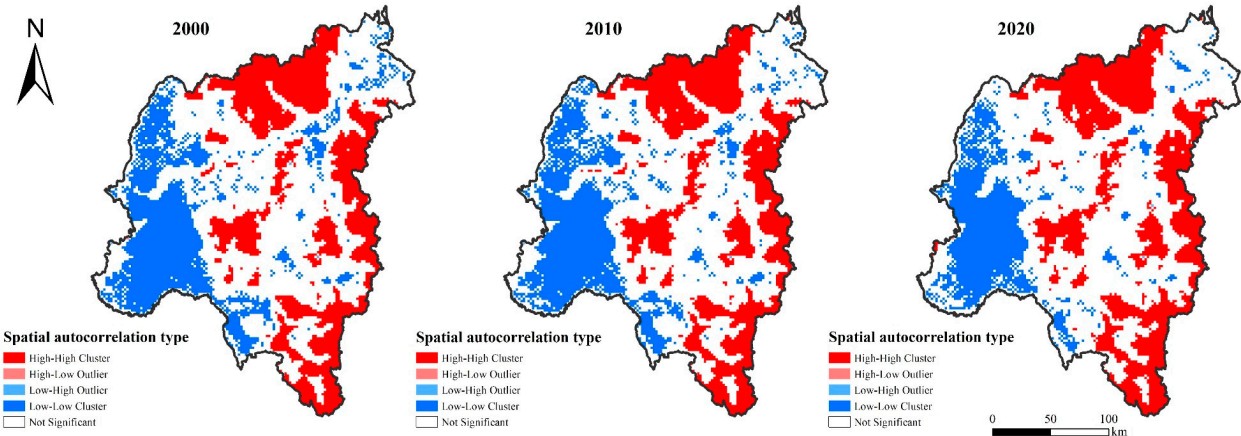

**Figure 6.** Spatial differentiation of habitat quality in Harbin from 2000 to 2020.

### 3.3. Result of Identification of Driving Factors

The influence levels of natural and social indicators were computed in this study using the geographic detector with factor detection and interaction detection to evaluate the drivers of habitat quality in Harbin City (Table 7, Figure 7). Seven driving factors were selected for factor detection: PRE, TMP, DEM, SLO, NDVI, POP, and GDP. The average q-values were ranked as POP (0.524) > NDVI (0.454) > GDP (0.436) > DEM (0.400) > TMP (0.335) > PRE (0.282) > SLO (0.177). Socio-economic elements had the most impact on the condition of the habitat, and the geographical differential features of habitat quality were strongly impacted by human activity. Strong explanatory power was also demonstrated by NDVI and DEM for habitat quality. The least effective explanatory factor for habitat quality was slope.

**Table 7.** Explanatory power of single factor on the spatial heterogeneity of habitat quality.

| Year | Climate Factors | | Topographic Factors | | Vegetation Factors | Human Factors | |
|---|---|---|---|---|---|---|---|
| | **PRE** | **TMP** | **DEM** | **SLO** | **NDVI** | **POP** | **GDP** |
| 2000 | 0.316 | 0.341 | 0.405 | 0.178 | 0.533 | 0.506 | 0.427 |
| 2010 | 0.230 | 0.338 | 0.398 | 0.179 | 0.438 | 0.532 | 0.424 |
| 2020 | 0.301 | 0.326 | 0.398 | 0.174 | 0.390 | 0.534 | 0.457 |

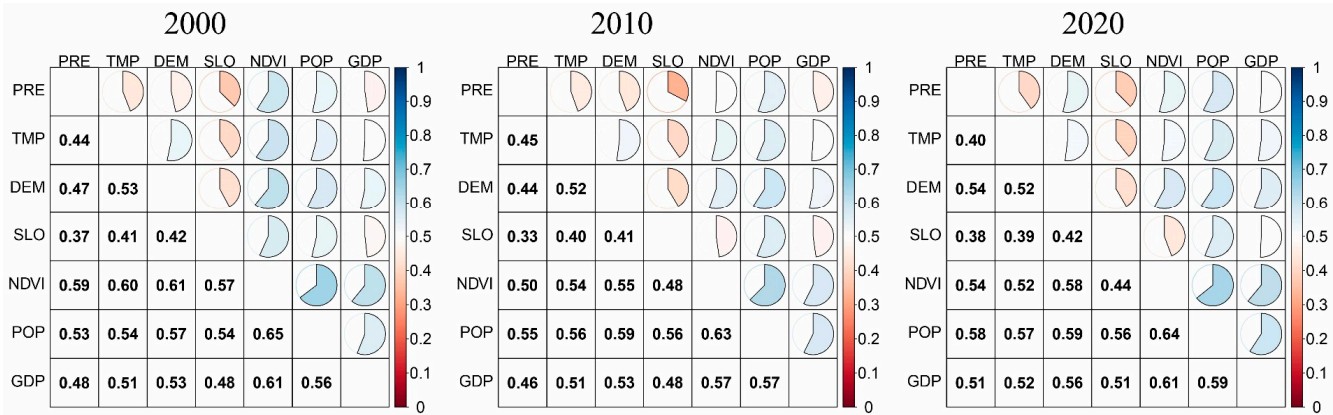

**Figure 7.** Results of the habitat quality driving factor interaction detection from 2000 to 2020.

To identify the impacts of many variables operating in concert with habitat quality, this study employed interaction detection. POP and NDVI had the greatest interaction explanatory power values, both above 0.6. Interaction detection revealed a robust explanatory power between NDVI, POP, GDP, and other factors, establishing them as crucial driving forces influencing habitat quality. In contrast to factor detection, interaction detection exhibited an enhanced explanatory power for habitat quality, suggesting that factor interactions significantly influenced habitat quality to varying degrees.

### 3.4. Spatial Interactions among Driving Factors and Habitat Quality

A multicollinearity test for each driver was conducted before performing GWR, and all drivers passed the multicollinearity test. Analyzing how social and ecological variables affect habitat quality using GWR (Figure 8), the local R$^2$ of the model was generally high, and the standard residual values were in the range of −2.5 to 2.5, which indicated a better fit and reliable results.

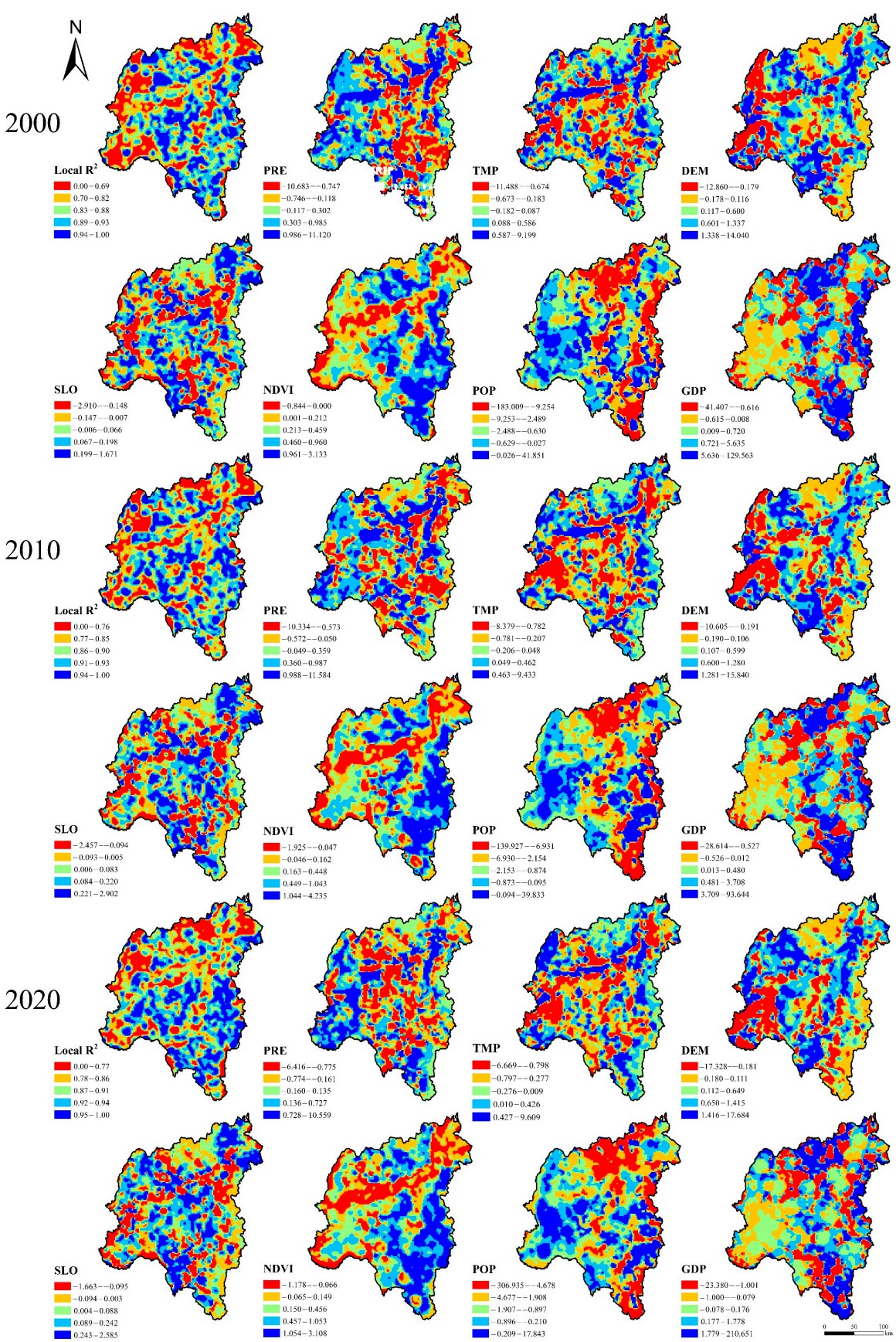

**Figure 8.** The quantitative effects of landscape indices in depicting habitat quality through GWR from 2000 to 2020.

Precipitation and temperature had close to 50% of the area positively affecting habitat quality, while DEM had more than 70% of the area positively affecting habitat quality. SLO had over 55% of the area positively impacting habitat quality. In the western region, the terrain is predominantly flat, featuring gentle slopes, increased human activities, and a prevalence of LULC types dominated by construction and agricultural land. This combination contributes to the degradation of habitat quality in the area. In higher-elevation hilly areas, the terrain is more undulating and there is less human disturbance; thus, the habitat quality is higher. NDVI positively affected habitat quality in more than 70% of the area, and the negative impacts were mainly along the Songhua River in a linear distribution. An increase in the NDVI results in increased plant cover, decreased human activity, less influence on the biological environment, greater landscape connectedness, and, ultimately, higher habitat quality. In more than 80% of the locations, POP had a detrimental influence on the quality of the habitat, and the area affected by POP showed a tendency toward continual expansion. This indicates that the stronger the human disturbance, the higher the ecological damage and the lower the habitat quality. On the other hand, GDP had more than 50% of its area positively impacting habitat quality, but the negatively impacted area in 2020 was larger than the negatively impacted area 20 years earlier in 2000.

### 3.5. Simulation of LULC

Based on the previous LULC data, the LULC situation in Harbin City in 2030, 2040, and 2050 was simulated and predicted (Table 8, Figure 9). The results show that construction land in Harbin is growing quickly and is mostly located in the western urban region. Concurrently, there is a continual reduction in agricultural land and grassland areas. However, there is an observable increase in the extent of forested areas, wetlands, and water bodies. The proportion of bare land area is still the smallest. From 2030 to 2050, there will be a trend of simultaneous advancement of social and economic development and ecological protection.

**Table 8.** LULC area and percentage of study area from 2030 to 2050.

| Types | 2030 | | 2040 | | 2050 | |
|---|---|---|---|---|---|---|
| | Area/km$^2$ | Proportion/% | Area/km$^2$ | Proportion/% | Area/km$^2$ | Proportion/% |
| Agricultural land | 24,828.05 | 46.79 | 23,989.40 | 45.21 | 23,270.71 | 43.85 |
| Forest | 19,259.47 | 36.30 | 19,439.21 | 36.63 | 19,576.18 | 36.89 |
| Grassland | 4056.10 | 7.64 | 3974.47 | 7.49 | 3926.33 | 7.40 |
| Wetland | 1030.82 | 1.94 | 1129.75 | 2.13 | 1209.37 | 2.28 |
| Water body | 1137.66 | 2.14 | 1337.08 | 2.52 | 1519.00 | 2.86 |
| Construction land | 2728.15 | 5.14 | 3171.02 | 5.98 | 3539.26 | 6.67 |
| Bare land | 22.76 | 0.04 | 22.09 | 0.04 | 22.16 | 0.04 |

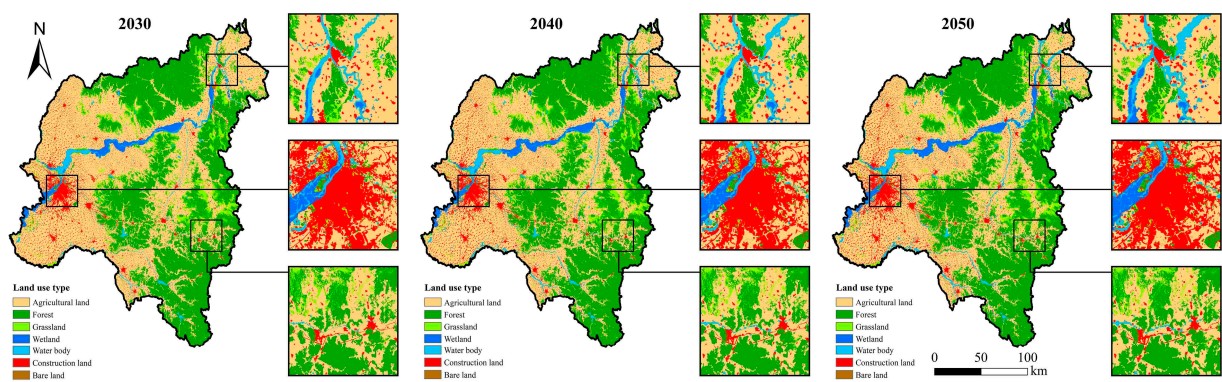

**Figure 9.** Map of future simulated LULC types from 2030 to 2050.

### 3.6. Simulation of Habitat Quality

The area percentage of each habitat quality category between 2030 and 2050 (Table 9) and the spatial distribution pattern (Figure 10) show that the overall habitat quality of Harbin City will decline from 2030 to 2050. The average habitat quality value is 0.7207 in 2030, 0.7152 in 2040, and 0.7113 in 2050. The highest proportion of the area is still a medium-quality habitat, but it continues to decrease. The area proportions of low-, low–medium-, medium–high-, and high-quality habitats all show a trend of increasing. From a spatial distribution standpoint, the low-quality habitat area in the western part of the city is anticipated to undergo further expansion, aligning with the observed trend of increased construction land. Concurrently, the habitat in the northeastern portion of the city shows significant improvement, corresponding to the expansion of water body areas. Additionally, as the forested area grows, the eastern portion sees an increase in high-quality habitat area.

**Table 9.** The area and proportion of habitat quality at different levels from 2030 to 2050.

| Levels | 2030 | | 2040 | | 2050 | |
|---|---|---|---|---|---|---|
| | Area/km$^2$ | Proportion/% | Area/km$^2$ | Proportion/% | Area/km$^2$ | Proportion/% |
| Low | 2813.22 | 5.30 | 3288.02 | 6.20 | 3675.97 | 6.93 |
| Low–medium | 881.14 | 1.66 | 1037.49 | 1.96 | 1159.56 | 2.19 |
| Medium | 24,486.50 | 46.15 | 23,449.70 | 44.19 | 22,589.64 | 42.57 |
| Medium–high | 1631.04 | 3.07 | 1784.12 | 3.36 | 1835.30 | 3.46 |
| High | 23,251.11 | 43.82 | 23,503.68 | 44.29 | 23,802.55 | 44.86 |

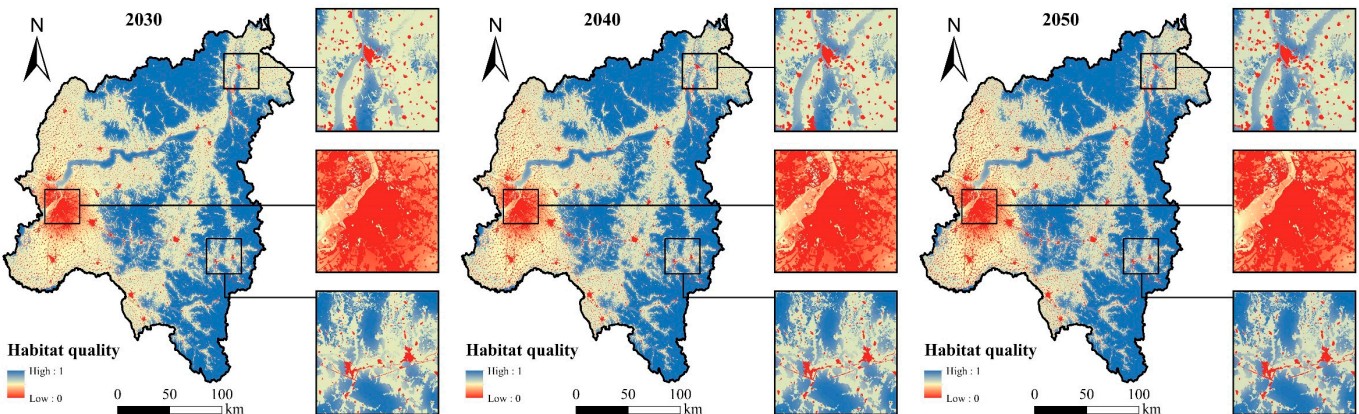

**Figure 10.** Spatial distribution pattern of habitat quality from 2030 to 2050.

## 4. Discussion

### 4.1. Response of Habitat Quality to LULC Change

Landscape ecology study has long focused on the link between ecological systems and landscape patterns [43,44]. The primary factors causing changes in habitat quality are LULC alterations [45]. The effect of LULC change on this region's habitat quality is a significant and intricate problem. Harbin's industrialization and urbanization have advanced further as a result of the country's reform and opening-up policies as well as the resuscitation of the former industrial base in Northeast China [46,47]. Northeast China's construction land area has grown quickly, and a sizable portion of natural land has been transformed into urban centers. The increase in LULC intensity has drastically damaged urban habitats, and the extension of habitats of worse quality has also been trending to increase yearly. The main areas of higher-quality habitat were the wetland habitat along the Songhua River and the heavily wooded areas of the Xiaoxing'an Mountains and Zhangguangcai Range. The higher-quality habitat area is growing along with the most natural wetland and forest areas, which may be connected to the many wetland and forest preservation strategies that cities have implemented. To stop the depletion of wetland resources, Harbin launched

special campaigns and the "Green Shield" program in recent years. It has also planned an ecological corridor spanning 100 miles along the Songhua River, safeguarded wetland parks and nature reserves, and significantly enhanced the quality of the habitat.

It was predicted that between 2030 and 2050, the areas of forest, wetland, and water bodies will all increase, which reflects that Harbin City is implementing ecological protection policies along with socio-economic progress. However, the overall habitat quality is forecasted to decline from 2030 to 2050, which is related to the rapid expansion of urbanization, especially the built-up areas in the western part of the city that were concentrated in patches, greatly reducing the level of habitat quality.

Areas dominated by forest have a high level of habitat quality in Harbin, while habitat quality is relatively low in densely populated, built-up areas, a finding consistent with previous studies [20,48,49]. In general, forest, grassland and water bodies are the areas in which high values of habitat quality are concentrated, while settlements, sandy areas, and bare land have the lowest habitat quality [21,50,51]. Although the study area is different and the climate, topography, and natural resources vary considerably from region to region, the conclusion that natural ecosystems have a better ecological environment compared to urban ecosystems is generally applicable. In summary, land use change directly affects the evolution of habitat quality. We must carefully weigh the importance of ecosystem and habitat quality to regional sustainable development to accomplish integrated socio-economic and ecological system development. In terms of LULC planning, it is important to logically distribute different LULC types, increase land preservation and restoration, and advocate for sustainable LULC practices [52].

### 4.2. Methodological Considerations

In this study, a combination of multiple models was used to make the study richer and more scientific. Based on the InVEST and PLUS models, the future development of habitats was simulated based on the analysis of historical habitat quality, and the spatial and temporal evolution characteristics of habitat quality in Harbin City were assessed for the period of 2000–2050. And the degrees of influence of natural and social factors on the spatial heterogeneity of habitat quality were spatialized by combining the geodetector and geographically weighted regression model to provide a scientific reference for future urban planning. This methodological linkage of multiple models and the presentation of site-specific spatial management recommendations are the novelties of this paper.

The degree of habitat quality was spatialized mainly using the InVEST model. Compared with field surveys used to obtain habitat parameters, this method has the advantages of easy data availability, simple operation and wide applicability [53]; these qualities are suitable for large-scale spatial planning, but the method also has certain uncertainties. In this study, habitat types were mainly classified according to land use types, which is a common classification method [20,28]. However, the spatial distribution pattern of habitat quality can vary considerably depending on the method of habitat classification. EUNIS (European Nature Information System) provides a detailed classification of different habitat suitability, and researchers have provided more detailed secondary classifications of forest types in Russia and North America [54]. Robert D. Pfister and other researchers describe a methodology for classifying forest habitat types based on potential climax vegetation through field research [55]. This categorization is more accurate compared to large-scale spatial model data, but the data are also more difficult to obtain. Celina Aznarez classified the habitats of Vitoria-Gasteiz into 18 classes according to their natural value, with a detailed delimitation of the different land classes. Level 1 represents completely sealed areas (buildings, bus lanes, highways, pavement and asphalt paths, etc.) and Level 18 represents land cover, presumably under the least amount of human influence (high mountains, lakes, and streams). And the habitat suitability score was determined via structured research with 21 experts [56], which made the model output more scientific. Instead of categorizing habitats, habitat grading is used to classify habitats with similar natural values into one level, and this method of grading habitats based on their natural values is also worthwhile.

Compared with the generalized land-use classification, it is more detailed, and compared with the complex habitat classification of field research, it is more concise and easier to calculate.

### 4.3. Suggestions for Land Space Optimization Based on Habitat Quality Improvement

The western region of Harbin City is primarily an agricultural area with a high concentration of human activities, a high level of urbanization, and nutrient-rich black soil [57]. Construction land is also expanding, with serious industrial and agricultural pollution. In addition, soil erosion has had a significant negative influence on the region's ability to produce food due to decreasing soil fertility and habitat quality loss, both of which have an adverse effect on human well-being. The ecological barrier areas of the Zhangguangcai Range in the east and center of the region and the Xiaoxing'an Mountains in the north both have higher levels of biodiversity, less human involvement, greater ecosystem services, and more forest cover. Therefore, priority locations for habitat should be selected, and various ecological restoration techniques should be chosen based on local characteristics, including the existing state of LULC as well as potential future scenarios. Throughout the city's western sector, the focus is on comprehensive soil erosion prevention and control projects, and agricultural farming measures, forestry, and grassland measures are combined with engineering measures to protect rare black soil resources. Near the Songhua River, it is recommended to implement water pollution control initiatives alongside wetland development, protection, and management projects. These efforts should prioritize the control of domestic sewage discharge and minimize pesticide and fertilizer loss, emphasizing the unique value of wetlands and the overall protection of wetland ecosystems. The establishment of ecological protection and restoration belts inside the ecological barrier zones of the Xiaoxing'an Mountains and Zhangguangcai Range can make a substantial contribution to the preservation and rehabilitation of local ecosystems.

Overall, although the connectivity of habitats in Harbin is insufficient, especially in the western region, the high degree of urbanization development and large areas of agricultural land have meant that higher-quality habitats are relatively scarce and more fragmented than in the east. In addition, the large amount of industrial and agricultural production and construction have caused water pollution and soil erosion, which is in strong contrast to the ecologically sound north and east. Therefore, it is recommended to use linear elements such as rivers and greenways as corridors to connect nature reserves and maintain biodiversity. In particular, the green corridor in the west of the city can be extended to connect it with the east. For the Songhua River to continue supporting national parks and nature reserves, as well as the Xiaoxing'an Mountains and Zhangguangcai Range's ecological barrier zones, these areas must be preserved for them to collectively carry out the crucial ecological roles of ecological corridors.

### 4.4. Limitations and Outlook

The InVEST model is a mature, well-developed model, characterized by easy access to data and its ease of operation [53]. However, the criteria used in its computation are rather subjective; thus, more research is needed to determine whether the parameters are accurate and logical. This research investigated how socio-economic and ecological variables affect habitat quality, but because the city is a sophisticated social-ecological system, how hidden elements like culture and policy affect habitat quality should also be considered. Therefore, future research must take into account additional affecting elements and describe how they interact.

To create a scientific foundation for future habitat quality development, this study examined how habitat quality will evolve in 2030, 2040, and 2050 under the inertial development scenario. However, it is noteworthy that the study did not include a comparative analysis of habitat quality differences among the various development scenarios.

Future research should consider the extent to which additional factors influence habitat quality and simulate the geographical distribution of habitat quality under various

situations to provide a thorough study of habitat quality under various scenarios and to provide more in-depth insights for urban ecological planning. Specifically, the ecological protection scenario, economic priority scenario, and agricultural land protection scenario could be considered for a thorough comparative evaluation of similarities and differences in habitat quality.

## 5. Conclusions

Employing both the InVEST and PLUS models, this study conducted a comprehensive analysis of the regional and temporal changes in habitat quality in Harbin City spanning from 2000 to 2050. As a complement, utilizing geographic detector and geographically weighted regression, we investigated how social and environmental variables influence habitat quality. There are four major findings of the study:

(1) Agricultural land and forest were the main LULC categories in Harbin City from 2000 to 2020. Large tracts of agricultural land were transformed into forests and building sites as a result of the combined effects of increasing urbanization and the ongoing implementation of ecological protection laws. The built-up area of the city expanded through encroachment into agricultural land. The heterogeneity of the landscape in Harbin City continued to increase from 2000 to 2020, the degree of fragmentation decreased, and the degree of human interference generally showed a decreasing trend.

(2) The habitat quality index of Harbin City exhibited stability around 0.72 from 2000 to 2020, with a slight upward trend. In spatial distribution, the prevailing pattern displayed a gradient of habitat quality, characterized by lower quality in the west and higher quality in the east. Around 50% of the city's area comprised medium-quality habitat, while approximately 40% constituted high-quality habitat, demonstrating a consistent upward trajectory. The research area's habitat quality is generally excellent, suggesting a promising course for growth.

(3) The most explanatory power for habitat quality was found in population density; nonetheless, over 80% of the area had detrimental effects on habitat quality. The slope had less of an impact on habitat quality than NDVI, GDP, and elevation, although all three demonstrated significant explanatory power. Habitat quality is typically positively impacted by natural factors.

(4) Harbin will concurrently achieve socioeconomic development and environmental preservation from 2030 to 2050. However, the overall habitat quality continues to decline.

**Author Contributions:** Conceptualization: Y.Q. and Y.H.; data curation, formal analysis, writing—original draft and writing—review and editing: Y.Q.; funding acquisition: Y.H. All authors have read and agreed to the published version of the manuscript.

**Funding:** This research was funded by the Heilongjiang Provincial Key R&D Program Project (CN), grant number GZ20220117.

**Data Availability Statement:** The data presented in this study are available on request from the corresponding author. The data are not publicly available due to the nature of this research.

**Conflicts of Interest:** The authors declare no conflict of interest.

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
