# Peer review of "Spatiotemporal Variation and Driving Factors Analysis of Habitat Quality: A Case Study in Harbin, China"

_land, doi:10.3390/land13010067_

Round 1
Reviewer 1 Report
Comments and Suggestions for Authors
Assessment of the quality of the habitat and the forecast of its dynamics are urgent problems. The topic of the paper is relevant and interesting for readers.
However, the interest for readers should be increased. Now the paper is interesting only for China. I advise the authors to make a comparison with the situation on this issue in other countries.
The problem of ecological classifications of habitats and vegetation can also be discussed. For example, compare the land classification used by the authors and Habitat type classifications developed in Western United States [Pfister, R.D.; Arno, S.F. Classification of forest habitat types based on potential climax vegetation. For. Sci. 1980, 26, 52-70], EUNIS habitat classification designed for Europe [EUNIS habitat classification. Updated in January 2023. Available at: https://www.eea.europa.eu/data-and-maps/data/eunis-habitat-classification-1/eunis-terrestrial-habitat-classification-review-2021]
I recommend the authors to familiarize themselves with the paper devoted to this topic. Ivanova, N.; Fomin, V.; Kusbach, A. Experience of Forest Ecological Classification in Assessment of Vegetation Dynamics. Sustainability 2022, 14, 3384. https://doi.org/10.3390/su14063384
A comparative analysis of the scientific approaches listed above and the approaches and methods chosen by the authors will make the paper more understandable and interesting for a wide range of readers living in different countries and climatic zones.
Also, from this point of view, it is desirable to strengthen the discussion.
This revision will also be useful to increase the originality of the study and its scientific and practical significance.
The methodology approaches are described in detail. The authors used methods adequate to the tasks set.
The research results are illustrated with figures and tables that are informative and do not duplicate each other. The paper contains 8 informative tables and 10 visual figures. The results are presented clearly and clearly.
The abundance of abbreviations complicates the perception of the results. It is advisable to solve this problem.
The Discussion section is an important part of the paper. It is here that the authors should determine the place of their research in world science, compare the results obtained with those of other studies, including those carried out in other countries and climatic zones, explain the strengths of the approaches used, the novelty, the practical and theoretical significance of the results. This is currently not the case. It is therefore necessary to improve this section.
Conclusions follow from the results and are reasonable. The paper will be of interest to a wide range of readers whose scientific interests are related to habitat quality. Despite the fact that English is not my native language, I read the paper with interest and had no difficulties in understanding. The paper corresponds to the subject of Land.
Author Response
Dear Reviewer,
We feel great thanks for your professional review work on our article. Thank you very much for reviewing our manuscript entitled “Spatiotemporal Variation and Driving Factors Analysis of Habitat Quality: A Case Study in Harbin, China”. All of these comments were valuable and helpful in improving our article. Based on the comments, we have made some changes to the manuscript and added additional data notes to make our results more convincing. In this revised version, our changes to the manuscript are marked in yellow text. Detailed point-by-point responses are provided below.
1. Now the paper is interesting only for China. I advise the authors to make a comparison with the situation on this issue in other countries.
(1) The problem of ecological classifications of habitats and vegetation can be discussed. For example, compare the land classification used by the authors and Habitat type classifications developed in Western United States [Pfister, R.D.; Arno, S.F. Classification of forest habitat types based on potential climax vegetation. For. Sci. 1980, 26, 52-70], EUNIS habitat classification designed for Europe [EUNIS habitat classification. Updated in January 2023. Available at: https://www.eea.europa.eu/data-and-maps/data/eunis-habitat-classification-1/eunis-terrestrial-habitat-classification-review-2021]
(2) I recommend the authors to familiarize themselves with the paper devoted to this topic. Ivanova, N.; Fomin, V.; Kusbach, A. Experience of Forest Ecological Classification in Assessment of Vegetation Dynamics. Sustainability 2022, 14, 3384. https://doi.org/10.3390/su14063384
Reply: Thanks for your review.
We have added the chapter [4.2. Methodological Considerations], which is dedicated to exploring how this study's habitat classification methodology compares with other researchers, including the papers by Pfister, R.D. and Ivanova, N. that you mentioned, as well as the EUNIS dataset.
This chapter explores the InVEST model habitat classification system, choosing Celina Aznarez's thesis as an example of the similarities and differences in methodology, comparing the two methods of habitat classification and habitat categorization, combined with the categorization of habitat field studies.(L839-L871)
(3) The Discussion section is an important part of the paper. It is here that the authors should determine the place of their research in world science, compare the results obtained with those of other studies, including those carried out in other countries and climatic zones, explain the strengths of the approaches used, the novelty, the practical and theoretical significance of the results.
Reply: Thank you very much for your suggestions.
We compare the results of this study with Ebinur Lake, Lithuania, Beressa watershed, and Pico Island based on your recommendations and conclude that habitat quality in natural ecosystems is generally higher than habitat quality in urban ecosystems. Also summarized are the major land classes where high and low habitat quality distribution areas are located. Specific descriptions have been added within the paper. (L820-L838)
We summarized the advantages of combining multiple models in this study, analyzing the evolution of historical habitat quality in Harbin with the help of the InVEST-PLUS model, and predicting future habitat development. With the help of the GD-GWR model, the degree of influence of ecological and social factors on habitat quality is explored and spatialized. Specific descriptions have been added within the paper. (L840-848)
2. The abundance of abbreviations complicates the perception of the results. It is advisable to solve this problem.
Reply: Thanks for your review. We avoided this problem by changing the abbreviation of the landscape pattern indexes to their full names.
These are our responses to the review comments and thank you for your suggestions to make our paper better.
Reviewer 2 Report
Comments and Suggestions for Authors
I've gone over the authors' responses and the changes to the text again. Now, they are already corresponding to each other. The replies and the text edits are handled correctly. The methodology of the paper has become clearer.
One last minor comment should be stated and justified in the text.
Why does the moving window used in calculating the landscape metric have a size of 720 m? When the image used has a resolution of 30m/px. I would expect a basic kernel of 3x3, possibly 5x2 (it's 90x90m / 150x150m, but 720m)?!.
Such a kernel is too smoothing!
Rating: minor revision.
Author Response
Dear Reviewer,
We feel great thanks for your professional review work on our article. Thank you very much for reviewing our manuscript entitled “Spatiotemporal Variation and Driving Factors Analysis of Habitat Quality: A Case Study in Harbin, China”. The comment was valuable and helpful in improving our article. Based on the comments, we have made some changes to the manuscript. In this revised version, our changes to the manuscript are marked in yellow text. Detailed responses are provided below.
Why does the moving window used in calculating the landscape metric have a size of 720 m?
Reply: Thanks for your review. We spatialize the landscape pattern indices through the moving window method, where the radius of the moving window is critical. A moving window scale that is too small leads to discontinuity in the generated image, while a window that is too large leads to loss of overall image detail and blurring of the generated image. In this study, a multiple of 30m was used as the radius of the moving window, and the optimal radius was determined by calculating the nugget-sill ratio. When the moving window radius is 720m, the nugget-sill ratio tends to be stable and less fluctuating, indicating that the optimum has been reached. Specific descriptions have been added within the paper. (L325-L332)
These are our responses to the review comments and thank you for your suggestions to make our paper better.
Reviewer 3 Report
Comments and Suggestions for Authors
This article primarily investigates the spatiotemporal changes in habitat quality in Harbin, China, and analyzes its driving factors. A comprehensive study is conducted on the evolving patterns from 2030 to 2050 under the scenario of inertia development, proposing spatial optimization strategies. Overall, while the article contributes to the research on habitat quality changes and influencing factors, there are still some deficiencies that need further improvement and refinement.
1. In Line 93, the section introducing the research area should supplement information on local climate conditions, such as annual temperature, annual precipitation, and descriptions related to habitat quality, as well as key elements highlighting the significance of the ecological environment in the region.
2. In Line 104, please provide details on the temporal and spatial resolution of the adopted data.
3. In Line 105, there is an abbreviation error ("LULU" instead of "LULC"). Throughout the entire article, it should be corrected to "LULC."
4. In Line 120, remove the ‘.’ after "types" as it disrupts the formatting.
5. In Line 132, please provide a detailed explanation of the calculation method for Dxy. This is crucial for the calculation of habitat quality in the article and should not be ambiguous. Additionally, specify the source of the method used at this point in the text.
6. All the images are unclear, and the quality needs improvement, especially for figures like Figure 2 and Figure 4, where it is difficult to discern the text on the images.
7. Consideration of factors is insufficient: The article mainly considers the impact of socio-economic and ecological variables on habitat quality. However, the city is a complex socio-ecological system, and factors such as culture and policies should also be considered in their impact on habitat quality.
8. Please ask a native speaker to polish language.
Comments on the Quality of English LanguagePlease polish the English
Author Response
Dear Reviewer,
We feel great thanks for your professional review work on our article. Thank you very much for reviewing our manuscript entitled “Spatiotemporal Variation and Driving Factors Analysis of Habitat Quality: A Case Study in Harbin, China”. All of these comments were valuable and helpful in improving our article. Based on the comments, we have made some changes to the manuscript and added additional data notes to make our results more convincing. In this revised version, our changes to the manuscript are marked in yellow text. Detailed point-by-point responses are provided below.
1. In Line 93, the section introducing the research area should supplement information on local climate conditions, such as annual temperature, annual precipitation, and descriptions related to habitat quality, as well as key elements highlighting the significance of the ecological environment in the region.
Reply: Thank you very much for your suggestions. Information on mean annual temperature, mean annual precipitation, and descriptions related to habitat are added within the paper. Specific descriptions have been added within the paper. (L257-L266)
2. In Line 104, please provide details on the temporal and spatial resolution of the adopted data.
Reply: Thanks for your review. We changed the text in the [2.2 Data Sources] section to a tabular format, adding information such as spatial and temporal resolution. (L269-L271)
3. In Line 105, there is an abbreviation error ("LULU" instead of "LULC"). Throughout the entire article, it should be corrected to "LULC."
Reply: Thank you for your suggestions. This was a mistake on our part and we have fixed the error here.
4. In Line 120, remove the ‘.’ after "types" as it disrupts the formatting.
Reply: Thank you for your suggestions. We have removed the "." , thank you very much for your careful correction.
5. In Line 132, please provide a detailed explanation of the calculation method for Dxy. This is crucial for the calculation of habitat quality in the article and should not be ambiguous. Additionally, specify the source of the method used at this point in the text.
Reply: Thank you very much for your suggestions. We have added detailed explanations of the formulas and added methodological sources. Specific descriptions have been added within the paper. (L336-L345)
6. All the images are unclear, and the quality needs improvement, especially for figures like Figure 2 and Figure 4, where it is difficult to discern the text on the images.
Reply: Thanks for your review. We have recreated the high-resolution images and enlarged the legends on the images as much as possible, making it possible for the reader to understand the content of the images.
7. Consideration of factors is insufficient: The article mainly considers the impact of socio-economic and ecological variables on habitat quality. However, the city is a complex socio-ecological system, and factors such as culture and policies should also be considered in their impact on habitat quality.
Reply: Thanks for your review. This is a point that we have not considered enough, and we have written about it in the [4.4. Limitations and Outlook] section. This is also the direction of our efforts for further research in the future, and we will consider the influencing factors of all aspects more comprehensively in our future research. (L933-L937, L942-L943)
8. Please ask a native speaker to polish language.
Reply: Thank you for your suggestions. We have checked and touched up the language throughout the text and made changes to address grammatical issues.
These are our responses to the review comments, and thank you for your suggestions to make our paper better.
Round 2
Reviewer 1 Report
Comments and Suggestions for Authors
The authors have responded to all my comments and improved the paper considerably. I have no further questions about the paper.